# Forgetting of task-specific knowledge in model merging-based continual learning

## Abstract

This paper investigates the linear merging of deep neural network models in the context of continual learning (CL). Using controlled visual cues in computer vision experiments, we demonstrate that merging largely preserves or enhances shared knowledge, while unshared task-specific knowledge rapidly degrades. We further find that merging models from an incremental training process consistently outperforms merging models trained in parallel.

## 1 Introduction

Methods that involve averaging the parameters of different models, often termed weight-space ensembling or model merging, have received significant attention in deep learning (Yang et al., 2024). A common application targets improving performance for a single task (Izmailov et al., 2018; Wortsman et al., 2022). When multiple (well-performing) model variants are trained starting from a common initialization – for instance, through multiple fine-tuning runs with different random seeds or different hyperparameter settings – averaging their weights tends to yield a final model with improved accuracy and robustness over any individual model, without increasing inference cost. This benefit is generally attributed to the geometry of the loss landscape, as models fine-tuned from the same initialization have been found to often reside in the same wide, and relatively flat basin (Goodfellow et al., 2015; Frankle & Carbin, 2019). A common interpretation is that averaging the weights of such models produces a solution closer to the center of this basin, smoothing out noise or over-specifications learned by individual training runs, leading to better generalization (Izmailov et al., 2018).

Beyond improving performance for single tasks, model merging has also been explored for integrating knowledge from multiple tasks into a single model (Ilharco et al., 2022; Matena & Raffel, 2022). This renders model merging a potential mechanism for continual learning (CL), where the goal is to learn multiple tasks sequentially without catastrophically forgetting earlier ones (McCloskey & Cohen, 1989; Parisi et al., 2019). With CL in mind, two approaches for merging models that are trained or adapted for different tasks can be distinguished. One option is to merge different states of an incrementally trained model, by averaging its weights after learning one task with its subsequent weights after learning a later task. Another option is to merge separate models trained in parallel on different tasks. Some studies have already shown promising results for model merging in specific CL contexts (Marouf et al., 2024; Udandarao et al., 2024; Kozal et al., 2024). Notably, the merging mechanism differs fundamentally from the typical approach to CL, which is to make changes to the loss function to sequentially approximate a joint objective over all observed tasks (Hess et al., 2023). Instead, weight-space merging is typically applied post-hoc, combining independently or sequentially trained models that have been specialized for individual tasks. The general conditions under which such post-hoc merging preserves or degrades knowledge across diverse tasks remain poorly understood. To better understand when merging is suitable for CL, we conceptualize the knowledge of a model trained on multiple tasks as comprising both *shared* components (e.g., general features from pre-training, or knowledge common across multiple tasks) and *task-specific* components (e.g., features or decision boundaries unique to a single task). The central question of our study then is: *how do these shared and task-specific knowledge components react during weight merging?*

In this work, we conduct controlled experiments targeting the computer vision domain. We propose a methodology that allows to instantiate either shared or task-specific knowledge via synthetic visual cues injected directly into the input image space. We empirically show that during linear weight interpolation, shared knowledge tends to be largely preserved or even enhanced, while unshared task-specific knowledge is significantly degraded. These results align with recent research by Zaman et al. (2024), who conduct related experiments with large language models. We further compare merging incrementally trained models with merging parallel trained ones. Our findings provide insight into the suitability and limitations of weight-space ensembling as a mechanism within various CL setups, potentially informing the design of more effective strategies for knowledge accumulation and retention.

## 2  Background

**Continual learning and model merging.** An important goal of continual learning is enabling models to learn sequentially from a stream of tasks, without catastrophically forgetting previously acquired knowledge (De Lange et al., 2022; Wang et al., 2024; van de Ven et al., 2025). The dominant approach in CL to incrementally training a single, evolving model involves attempting to constantly approximate a joint learning objective across all encountered tasks (Hess et al., 2023). Prominent examples of this approach include parameter regularization (Kirkpatrick et al., 2017), functional regularization (Li & Hoiem, 2017) and replay (Robins, 1995).

Model merging offers a different perspective, as it involves the post-hoc combination of models trained independently or sequentially. Indeed, simple averaging has shown promising results in the CL 'chunking' setting (Lee & Storkey, 2025), was combined with replay (Marouf et al., 2024), and has been explored in combination with other techniques for continuous adaptation, sometimes incorporating Fisher information or exponential moving average-like updates sequentially (Udandarao et al., 2024; Dziadzio et al., 2025). However, merging models adapted to different tasks still introduces interference (Yadav et al., 2023; Marczak et al., 2024), a key challenge shared with the stability-plasticity dilemma inherent to CL.

From the perspective of the loss landscape, this interference can be explained as models being adapted to distinct tasks converge to disparate local minima and are no longer connected by a linear path of low loss (Frankle & Carbin, 2019; Neyshabur et al., 2020). Consequently, averaging their weights can result in a solution that falls onto a high-loss barrier, degrading the specific capabilities acquired during adaptation. While model merging is often explored for its potential to aggregate capabilities in a model, the work of Zaman et al. (2024) investigated the inverse potential: using merging as a mechanism to selectively *forget* specific, potentially undesirable knowledge components, e.g. gender-bias, in LLMs. Our work investigates these dynamics in the vision domain to improve our understanding of the suitability of model merging for CL. We analyze interpolation across models trained with different task-specific cues to implicitly probe these geometric properties with respect to shared- and task-specific knowledge, and illustrate that depending on whether knowledge is shared between models or not, merging either preserves that knowledge or leads to interference.

**Cues, shortcuts, confounders.** To create distinct task-specific knowledge components in a controlled manner in the computer-vision domain, we draw inspiration from the literature on shortcut learning (Geirhos et al., 2020). This research area studies how deep neural network models are prone to exploiting spurious correlations instead of learning robust features. By injecting synthetic visual cues that are easy to learn and correlated with the class label, but distinct from the core image content, we encourage models to specialize on these particular 'shortcuts'. This setup allows for a cleaner separation between shared (same shortcut in each task) and task-specific (different shortcut in each task) knowledge compared to using standard datasets with semantic splits, where the nature of shared vs. specific features can be ambiguous (Hess et al., 2024). It provides a visual analog to the token-pair association used by Zaman et al. (2024) and allows us to study how merging interacts with models reliant on different, easily controlled, task-specific strategies. This controlled approach is also related to Busch et al. (2025), who study the impact of confounders to CL, but we focus specifically on the interaction of shared and task-specific knowledge when using merging-like approaches.

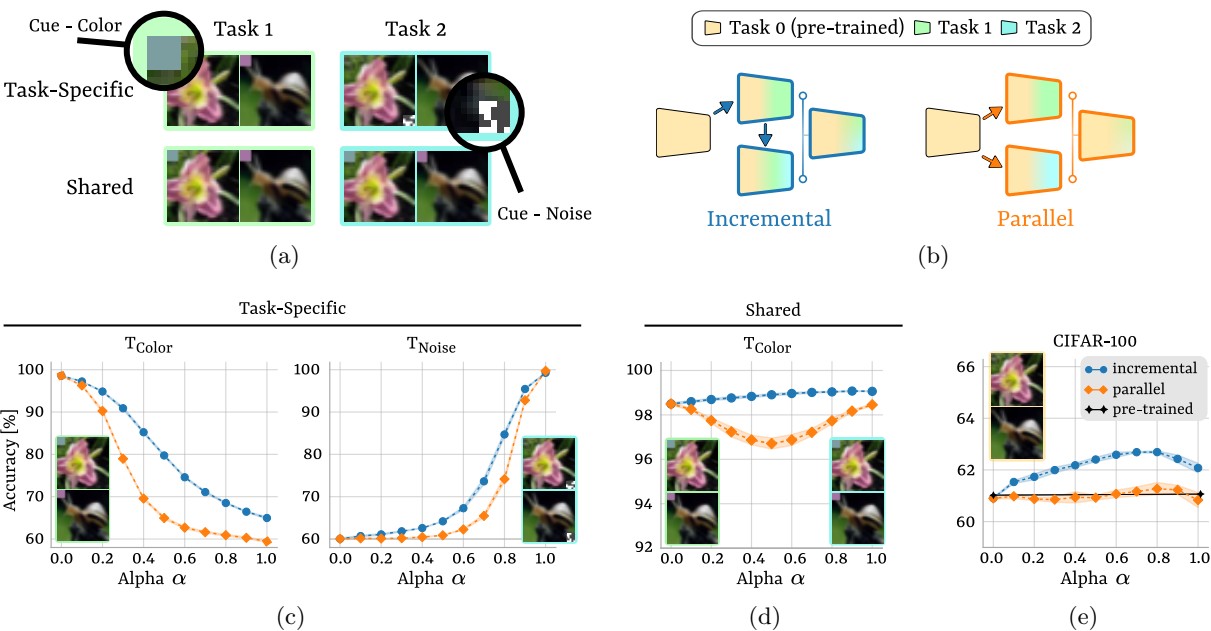

Figure 1: **Experimental Protocol and Results in the Standard Setup.** **(a)** Example of shared and task-specific knowledge instantiated with visual cues. The shared knowledge protocol uses the same cue for both tasks, while the task-specific knowledge protocol uses distinct cues. **(b)** Schematic illustration of 'incremental' (blue) and 'parallel' (orange) training. Both scenarios start from a common pre-trained model (Task 0, yellow) and adapt models for subsequent tasks (Task 1, green; Task 2, turquoise) that involve specific visual cues. **(c)** Accuracy (y-axis) vs. interpolation coefficient $\alpha$ (x-axis) for the task-specific knowledge protocol. Performance is evaluated in the presence of the color (left) and noise (right) visual cues, comparing incremental (blue circles) and parallel (orange diamonds) training. The endpoint models of the interpolation are specialized for $T_{\text{Color}}$ ($\alpha = 0$) and $T_{\text{Noise}}$ ($\alpha = 1$). **(d)** Accuracy vs. $\alpha$ for the shared knowledge protocol, where both endpoint models of the interpolation are specialized for $T_{\text{Color}}$. **(e)** Performance on the base dataset (no cues) when interpolating $T_{\text{Color}}$ ($\alpha = 0$) and $T_{\text{Noise}}$ ($\alpha = 1$) endpoints; the solid black line marks accuracy of the pre-trained model. The plotted lines in the bottom three panels represent the mean accuracy across three independent runs with different random seeds, while the shaded areas indicate the standard error. All evaluations are on the full CIFAR-100 test set. *These results show that while unshared task-specific knowledge degrades rapidly when models are merged (panel c), shared knowledge components are largely preserved or even enhanced (panel d,e). Moreover, merging incrementally trained models consistently leads to better knowledge preservation compared to merging models trained in parallel.*

## 3  Methodology

Our methodology is designed to investigate the effect on shared (common) and unshared (task-specific) knowledge during model merging in computer vision settings. To create controllable and distinct knowledge components in our models, we augment a base image dataset by superimposing synthetic visual cues onto the input images. Examples are shown in Figure 1a. A combination of the base dataset with a specific visual cue constitutes a distinct 'task' for the model to learn. Interpolating between endpoint models that learned tasks with either different or the same visual cues governs our investigation of task-specific and shared knowledge retention. For all our experiments, CIFAR-100 (Krizhevsky et al., 2009) is used as the base dataset. The full implementation details are presented in Appendix A.

**Shared pre-trained initialization.** All model trainings begin from a common base model, $\theta_{\text{PT}}$, which has been pre-trained on the base dataset. The primary purpose of pre-training is to ensure that all endpoint models share a significant initial learning trajectory, a condition motivated by linear mode connectivity research,

which suggests it facilitates meaningful weight-space merging without requiring complex re-parameterization techniques (Goodfellow et al., 2015; Frankle & Carbin, 2019; Ainsworth et al., 2023). In addition, the pre-training on the base dataset (without cues) yields a broad set of general features, which allows us to evaluate the preservation of this foundational 'shared knowledge' as an independent measure alongside our visual cue-specific evaluations, which we detail next.

**Visual cues.** We define a visual cue as an $N \times N$ pixel patch superimposed onto an image at a specific location, providing a simple visual pattern that is consistently correlated with the sample's ground-truth class label, irrespective of the underlying image content. To create two different types of shortcuts, we define two families of cues: (1) Colored patches, consisting of pixel patches of solid color (the *color cue*, denoted as $\mathcal{C}_{\text{color}}$). We utilize the HSV color space, setting Saturation (S) and Value (V) to 1.0, and distributing Hue (H) evenly across classes. (2) Grayscale noise patches, consisting of pixel patches containing grayscale noise patterns (the *noise cue*, $\mathcal{C}_{\text{noise}}$). For each class, a unique pixel noise pattern is generated by sampling each pixel's intensity independently and uniformly from the discrete set $\{0, 255\}$. For both families of cues, we use a patch size of $N = 5$. During training, a cue is superimposed on each image with a probability $p_{\mathcal{C}} = 0.5$ to incentivize models to learn both the cue and the general image features.

**Shared vs. task-specific knowledge.** We use the visual cues to construct two knowledge protocols that are at the core of our experimental setups. (1) The *unshared, task-specific knowledge protocol* consists of two tasks constructed using distinct visual cues: in one task the color cue is added to the base dataset (denoted as $T_{\text{color}}$), and in the other task the noise cue is added ($T_{\text{noise}}$). To minimize information transfer, the cues are placed at non-overlapping positions (top-left and bottom-right). (2) The *shared knowledge protocol* consists of two tasks with the same visual cue: in both tasks, the color cue is added to the base dataset at the same position (top-left). This controlled use of cues allows for a more explicit separation of shared versus unshared specific knowledge compared to typical CL benchmarks based on semantic splits, where disentangling preserved from re-discovered knowledge can be challenging (Hess et al., 2024).

**Incremental vs. parallel training.** Starting from the pre-trained weights $\theta_{\text{PT}}$, we train models on one of the knowledge protocols to generate *endpoint models* (i.e., models that are trained or adapted for a task with a particular visual cue) for our merging analysis. As illustrated in Figure 1b, we compare two distinct training scenarios. With incremental training, which represents continual learning without specific forgetting mitigating, factors like catastrophic forgetting or knowledge transfer between the sequential training stages can influence the characteristics of the endpoint models. In contrast, parallel training serves as a controlled baseline where both endpoint models are trained independently from the identical pre-trained state $\theta_{\text{PT}}$. This setup allows for a direct analysis of merging effects with and without sequential dependencies.

**Evaluation protocol.** We evaluate all models (pre-trained, endpoints, and interpolated) using classification accuracy on the held-out test set. To probe shared and task-specific knowledge induced by visual cues, we evaluate performance on the test set with either the color cue ($\mathcal{C}_{\text{color}}$) or the noise cue ($\mathcal{C}_{\text{noise}}$) deterministically ($p_C = 1.0$) applied. To measure the preservation of general shared knowledge, we evaluate performance on the original test set without any visual cues ($p_C = 0.0$) applied. Interpolated models $\bar{\theta}(\alpha) = \alpha \theta_{T_1} + (1 - \alpha) \theta_{T_2}$ are generated via linear interpolation of appropriate task-specific endpoints, with $\alpha \in [0, 1]$ the interpolation coefficient.

## 4 Results

Here we present the results of the merging experiments, with the endpoint models generated via either parallel or incremental training and according to either the shared or task-specific knowledge protocol.

### 4.1 Standard setup

The 'standard setup' resembles common model-merging setups, where the pre-trained model $\theta_{\text{PT}}$ establishes a strong foundation of general features on top of which further adaptation occurs. In our first set of experiments, the pre-trained model $\theta_{\text{PT}}$ is obtained from training on the full CIFAR-100 training set, establishing the shared general feature base before any cue adaptation. During cue adaptation, the full CIFAR-100 training set is used as well. This setup isolates the effect of merging on cue-specific knowledge, since the

shared knowledge base is already consolidated prior to the divergence into task-specific endpoints, and there is no forgetting of shared features in the adapted endpoints. Key trends are summarized in Figure 1, with full numerical results in Appendix C.

**Endpoint models successfully learn specific cues while retaining general knowledge.** Before investigating model interpolation, we confirm that the endpoint models behave as expected. Models adapted to a specific visual cue demonstrate high performance when tested on that cue, with cue-specific accuracies consistently above 98%, while performance on the shared general knowledge (CIFAR-100 without cues) remains close to the pre-trained baseline of 61%. An exception is the endpoint model incrementally trained on both cues, whose performance on CIFAR-100 without cues surpasses that of the pre-trained baseline, indicating positive transfer from the sequential training process.

**Unshared specific knowledge degrades while shared knowledge is preserved or enhanced.** A central finding of our work is the starkly different effect of linear interpolation on different knowledge types. As shown in Figure 1c, unshared task-specific knowledge rapidly degrades, i.e. sensitivity to a specific cue decays sharply as the interpolation moves towards the other endpoint, a transition that is particularly abrupt in the parallel setting. Conversely, *shared knowledge* is robust to interpolation. As shown in Figure 1d, when merging models that were trained on tasks with the same cue ($T_{\text{Color}}$), performance is maintained well across the interpolation path. Furthermore, as shown in Figure 1e, performance on the common CIFAR-100 task remains stable (in the parallel scenario) or is even enhanced (in the incremental scenario), peaking at an accuracy of 62.69%, which is above both endpoints.

**Merging incrementally trained models preserves knowledge better than merging parallel-trained ones.** We also find a key distinction between training scenarios, as merging models from an incremental training process consistently outperforms merging models trained in parallel (Figure 1c to 1e). In particular, when merging two models that are adapted for the same specific cue ($T_{\text{Color}}$), the incremental scenario maintains high performance across the entire interpolation path, whereas the parallel scenario exhibits a slight concave dip in accuracy around the midpoint (Figure 1d). Furthermore, for general shared knowledge (CIFAR-100 without cues, Figure 1e), the incremental scenario produces a notable synergistic effect: accuracy rises to a peak of 62.69%, surpassing both endpoints and the original pre-trained model. In contrast, the parallel scenario's performance remains largely flat, showing no significant benefit from interpolation. This finding, supported by related work of (Marouf et al., 2024), is particularly relevant for CL, as it indicates that merging states from a single, evolving model can be more advantageous than merging independently trained specialists.

## 4.2 Chunking setup

To endorse the generality of these observations, we test whether our findings hold when shared knowledge itself is acquired incrementally as well, rather than consolidated upfront as in the 'standard setup' above. We divide the CIFAR-100 training data into three disjoint chunks of approximately equal size, following the 'chunking' setup proposed by Lee & Storkey (2025). Pre-training is performed only on chunk 1. Subsequent cue adaptations occur on chunks 2 and 3, such that models must learn both general features and cue-specific knowledge simultaneously during adaptation. The methodology otherwise remains identical to Section 4.1.

**Results generalize to distributed learning of shared knowledge.** As shown in Figure 2, the core dynamics we observe remain consistent. Unshared task-specific knowledge (sensitivity to distinct cues) degrades rapidly during interpolation for both parallel and incremental scenarios. Shared general knowledge, now learned distributively across chunks rather than consolidated upfront, is preserved and enhanced through merging – with the incremental scenario again showing the largest gains. Notably, interpolated models in the incremental setting exceed the accuracy of both endpoints and the chunk-1 pre-trained baseline on CIFAR-100 without cues, demonstrating that merging can consolidate general knowledge even when it is acquired piecemeal across sequential training stages. These results suggest that the asymmetric treatment of shared versus task-specific knowledge under merging is robust to how shared knowledge is established, whether through comprehensive pre-training or incremental exposure across data partitions.

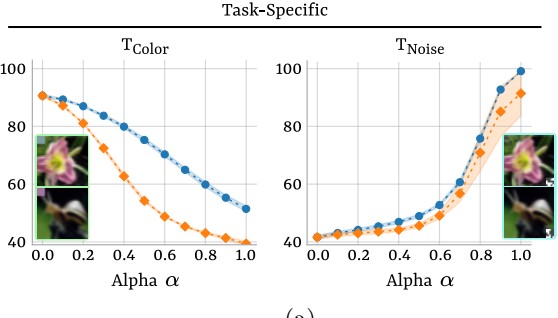
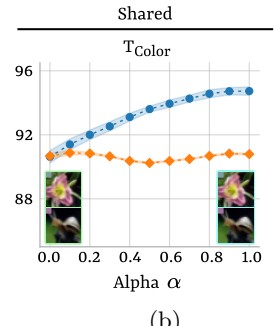
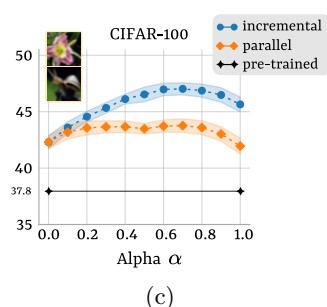

Figure 2: **Weight-Interpolation Results in the 'Chunking' Setup.** Pre-training uses only chunk-1 of CIFAR-100. Endpoint models are adapted on chunk-2 with $T_{\text{Color}}$ ($\alpha = 0$) and on chunk-3 with either $T_{\text{Noise}}$ or again $T_{\text{Color}}$ ($\alpha = 1$). **(a)** Task-specific knowledge protocol: accuracy on color (left) and noise (right) cues. **(b)** Shared knowledge protocol: accuracy on color when both endpoints were adapted with $T_{\text{Color}}$. **(c)** Performance on CIFAR-100 without cues, the black diamond marks pre-trained accuracy. Lines show mean accuracy across three runs and shaded areas indicate standard error. All evaluations are on the full CIFAR-100 test set. *The core dynamics from our first set of experiments persist: unshared task-specific knowledge degrades rapidly (panel a), while shared knowledge is preserved or enhanced (panel b,c), even when general features are learned distributively across chunks.*

## 5 Discussion

Our experiments demonstrate that linear merging of deep neural network models distinctly impacts different knowledge types. On one hand, shared knowledge is largely preserved and can be enhanced, aligning with earlier findings of consolidating commonalities within shared loss basins (Izmailov et al., 2018). On the other hand, unshared task-specific knowledge rapidly degrades upon interpolation due to interference between divergent parameter adaptations, a finding consistent with prior work on large language models (Zaman et al., 2024). Another important finding of our work is that merging incrementally trained models yields better knowledge consolidation than merging parallel-trained ones. This suggests that sequential adaptation, even without explicit CL mechanisms, guides models along trajectories that are more amenable to beneficial merging. This observation aligns with recent work by Marouf et al. (2024), who similarly find benefits from merging states of a continually evolving model rather than independently trained specialists.

Several factors may modulate the dynamics we observe. Our study used a ResNet-18, yet much recent merging success has been reported for very large models. As prior work (Ramasesh et al., 2022; Ilharco et al., 2022) suggests, the effectiveness of merging can improve with scale, possibly due to staying much closer to the pre-trained model weights, i.e. taking advantage of greater parameter capacity allowing for more sparse encoding of specific knowledge. Additionally, simple linear interpolation represents a baseline merging strategy. More advanced methods, such as Fisher-weighted averaging (Matena & Raffel, 2022) or interference-resolving techniques (Yadav et al., 2023; Marczak et al., 2024), might better preserve specific knowledge components. However, such methods primarily re-balance existing knowledge, meaning their utility still depends on whether preserving those specific components is desirable. Also the training regimen of endpoint models matters. Recent work on continual pre-training shows that specific optimization strategies, such as exponential moving averages, can significantly improve model suitability for subsequent merging (Udandarao et al., 2024; Marouf et al., 2024), pointing towards co-designing training and merging strategies.

For continual learning, our demonstrations highlight a critical balance: The degradation of task-specific knowledge during merging could be beneficial if such adaptations represent undesirable artifacts like biases or spurious correlations, as framed by Zaman et al. (2024). However, in many CL contexts, unique knowledge from past tasks is precisely what needs to be preserved. This suggests that simple merging is most suitable for CL scenarios aiming to consolidate general, shared capabilities, or where tasks are sufficiently similar so that their specific solutions remain compatible. For preserving diverse specialized knowledge, merging alone appears insufficient without tailored endpoint training or more sophisticated merging techniques. Opera-

tionally, linear merging remains attractive as a post-hoc, computationally inexpensive approach compared to other CL methods that modify training itself.

## 6 Limitations and future work

Our study offers insight into how linear merging of deep neural network models differentially affects shared and task-specific knowledge in the vision domain. While our controlled setting based on visual cues enabled clear distinctions, several avenues warrant further exploration to better understand the applicability of merging in continual learning. One is to investigate whether the observed dynamics of shared versus specific knowledge scale to larger, more diverse architectures (e.g., vision transformers) and more complex task sequences. While merging benefits are known for large models, the precise nature of knowledge interaction as identified here needs validation at scale. Another compelling direction for future work is to explore methods for achieving more explicit internal disentanglement of general versus task-specific knowledge within models during their (continuous) training.

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

## Appendix

The following Appendix provides supplementary material to accompany the main paper. It offers:

A Further details on our experimental setup

B Additional experimental validation

C Comprehensive numerical results for the primary experiments

## A Experimental setup details

This section provides further details on the experimental setup and hyperparameters used throughout all training phases (pre-training and subsequent cue adaptations) described in Section 3 and Appendix B, unless otherwise specified.

**Model architecture and optimization.** The core architecture is a slim ResNet-18 model (Lopez-Paz & Ranzato, 2017). In this model, batch normalization layers were replaced with group normalization layers using a single group to approximate layer normalization, thereby avoiding potential confounding effects from batch normalization's running statistics parameters (Kozal et al., 2024). All training stages (pre-training and subsequent cue adaptations) utilize a cross-entropy loss function and the same optimization settings. Optimization is performed using Stochastic Gradient Descent (SGD) with a momentum of 0.9 and weight decay of $5 \times 10^{-4}$. Each training stage proceeds for 50 epochs. A linear learning rate warm-up is applied during the first 5% of these epochs, increasing the learning rate from 0.004 to 0.1. Subsequently, a cosine annealing learning rate schedule decays the learning rate from 0.1 down to a minimum of $1 \times 10^{-5}$ over the remaining epochs. The mini-batch size is 128. All cue adaptation experiments are repeated three times with distinct random seeds, and reported as mean and standard error. Unless explicitly specified, pre-training is not subjected to different random seeds, but the same pre-trained weights are used as initialization for all runs.

**Data augmentation and visual cue application.** Standard data augmentations for CIFAR-100 are employed during all training stages: random horizontal flips (with a probability of 0.5) and random crops (image size $32 \times 32$ pixels, with padding of four pixels). Visual cues, when applied during the cue adaptation stages, are superimposed onto the images *after* these standard augmentations. For the visual cues (both $\mathcal{C}_{\text{color}}$ and $\mathcal{C}_{\text{noise}}$), we use a patch size of $5 \times 5$ pixels. When distinct cues are used to instantiate unshared task-specific knowledge (e.g., $T_{\text{Color}}$ vs. $T_{\text{Noise}}$ experiments), they are placed at fixed, non-overlapping, and distinct positions: the color patch in the top-left corner and the noise patch in the bottom-right corner of the image.

The code to reproduce our experiments is available at: https://anonymous.4open.science/r/MergeForget-BBBD/

## B Reverse order of cue adaptation

To further assess the robustness of our main findings regarding the interaction of shared and task-specific knowledge, we conducted an additional set of experiments where the order of cue adaptation in the incremental scenario was reversed compared to that primarily presented in Section 3.

**Experimental setup modification.** The core methodology for pre-training, visual cue design ($\mathcal{C}_{\text{color}}$ and $\mathcal{C}_{\text{noise}}$), optimization, and evaluation remains identical to our main experiments (details in Appendix A). The key difference lies in the reversed order of the cues in the task-specific adaptation phase.

**Results and observations.** The interpolation results for this reverse order experiment are presented in Figure 3 and Table 1. The observed trends are qualitatively consistent with those reported in our main results (Section 4. Unshared task-specific knowledge (sensitivity to the initial $T_{\text{Noise}}$ cue at $\alpha = 0$ or the final $T_{\text{Color}}$ cue at $\alpha = 1$) rapidly degrades when interpolating towards the opposing endpoint. Performance

Table 1: **Numerical Results for the Reverse Cue Order Experiments.** Shown for each experiment is the classification accuracy (in %) on the test set, reported as mean ± standard error across three runs with different random seeds.

| Training | Evaluation | Scenario | Alpha | | | | | | | | | | |
| | | | 0.0 | 0.1 | 0.2 | 0.3 | 0.4 | 0.5 | 0.6 | 0.7 | 0.8 | 0.9 | 1.0 |
|---|---|---|---|---|---|---|---|---|---|---|---|---|---|
| $T_{\text{Noise}} \to T_{\text{Color}}$ | Noise | *Incr.* | 99.76 ± 0.04 | 97.11 ± 0.33 | 87.44 ± 1.00 | 76.17 ± 0.99 | 69.57 ± 0.66 | 66.34 ± 0.37 | 64.68 ± 0.15 | 63.83 ± 0.04 | 63.36 ± 0.05 | 62.87 ± 0.18 | 62.39 ± 0.20 |
| | | *Para.* | 99.76 ± 0.04 | 93.70 ± 0.23 | 75.86 ± 0.14 | 66.39 ± 0.17 | 62.76 ± 0.06 | 61.31 ± 0.04 | 60.71 ± 0.06 | 60.51 ± 0.06 | 60.51 ± 0.12 | 60.48 ± 0.09 | 60.29 ± 0.12 |
| | Color | *Incr.* | 59.71 ± 0.12 | 60.70 ± 0.13 | 61.84 ± 0.02 | 63.03 ± 0.17 | 65.05 ± 0.23 | 68.59 ± 0.24 | 74.77 ± 0.28 | 83.52 ± 0.22 | 90.32 ± 0.05 | 94.25 ± 0.14 | 96.21 ± 0.13 |
| | | *Para.* | 59.71 ± 0.12 | 60.23 ± 0.10 | 60.60 ± 0.06 | 60.94 ± 0.04 | 61.73 ± 0.03 | 63.36 ± 0.08 | 67.26 ± 0.28 | 75.35 ± 0.49 | 87.22 ± 0.30 | 94.63 ± 0.26 | 97.59 ± 0.10 |
| | CIFAR-100 | *Incr.* | 61.07 ± 0.16 | 61.66 ± 0.13 | 61.98 ± 0.13 | 62.38 ± 0.15 | 62.68 ± 0.18 | 62.75 ± 0.16 | 62.82 ± 0.16 | 62.99 ± 0.20 | 63.12 ± 0.17 | 62.98 ± 0.11 | 62.81 ± 0.13 |
| | | *Para.* | 61.07 ± 0.16 | 61.35 ± 0.17 | 61.45 ± 0.20 | 61.43 ± 0.20 | 61.15 ± 0.13 | 61.02 ± 0.12 | 60.88 ± 0.08 | 60.94 ± 0.04 | 60.99 ± 0.06 | 61.14 ± 0.11 | 60.85 ± 0.13 |

on the shared CIFAR-100 task (no cues) again shows that the incremental adaptation scenario can lead to an enhancement of this knowledge around the midpoint of interpolation. The parallel scenario for shared knowledge remains relatively flat.

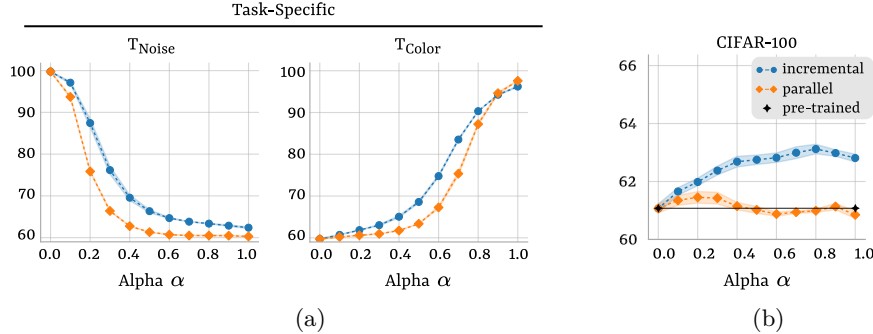

Figure 3: **Weight-Interpolation Results for Reverse Order Cue Adaptation.** Accuracy (y-axis) vs. interpolation coefficient $\alpha$ (x-axis). The $\alpha = 0$ endpoint is specialized on $T_{\text{Noise}}$ and the $\alpha = 1$ endpoint on $T_{\text{Color}}$. Results compare 'incremental' (blue circles) and 'parallel' (orange diamonds) training. **(a)** The task-specific panels ($T_{\text{Noise}}$, $T_{\text{Color}}$) show the performance on the respective cue-specific test sets. **(b)** The shared CIFAR-100 panel shows the performance without cues. In both panels, the plotted lines represent the mean accuracy across three independent runs with different random seeds, while the shaded areas indicate the standard error. All evaluations are on the full CIFAR-100 test set.

## C  Full numerical results

This section presents the comprehensive numerical data of the main experiments discussed in Section 4 and visualized in Figure 1. Table 2 details the mean classification accuracies and standard errors (across three runs) for all evaluated conditions: endpoint models ($\alpha = 0$ and $\alpha = 1$) and interpolated models ($\alpha \in (0, 1)$), trained in either the task-specific knowledge protocol ($T_{\text{Color}} \to T_{\text{Noise}}$) or the shared knowledge protocol ($T_{\text{Color}} \to T_{\text{Color}}$), for both parallel and incremental training scenarios, and evaluated with or without the visual cues used during training.

Table 2: **Numerical Results for the Standard Setup.** Displayed for each experiment is the test accuracy (in %), reported as the mean ± standard error across three runs with different random seeds.

| Training | Evaluation | Scenario | Interpolation Coefficient $\alpha$ | | | | | | | | | | |
| --- | --- | --- | --- | --- | --- | --- | --- | --- | --- | --- | --- | --- | --- |
| | | | 0.0 | 0.1 | 0.2 | 0.3 | 0.4 | 0.5 | 0.6 | 0.7 | 0.8 | 0.9 | 1.0 |
| $T_{\text{Color}} \to T_{\text{Noise}}$ | $T_{\text{Color}}$ | *Incr.* | 98.58 ± 0.05 | 97.22 ± 0.07 | 94.85 ± 0.13 | 90.89 ± 0.29 | 85.22 ± 0.29 | 79.74 ± 0.22 | 74.60 ± 0.20 | 71.08 ± 0.20 | 68.50 ± 0.19 | 66.46 ± 0.15 | 64.97 ± 0.24 |
| | | *Para.* | 98.57 ± 0.05 | 96.28 ± 0.10 | 90.25 ± 0.37 | 78.96 ± 0.15 | 69.56 ± 0.23 | 64.96 ± 0.27 | 62.69 ± 0.25 | 61.61 ± 0.22 | 60.92 ± 0.20 | 60.29 ± 0.19 | 59.43 ± 0.26 |
| | $T_{\text{Noise}}$ | *Incr.* | 60.05 ± 0.17 | 60.70 ± 0.15 | 61.11 ± 0.20 | 61.83 ± 0.10 | 62.59 ± 0.06 | 64.19 ± 0.11 | 67.27 ± 0.36 | 73.63 ± 0.75 | 84.70 ± 0.62 | 95.45 ± 0.26 | 99.26 ± 0.05 |
| | | *Para.* | 60.05 ± 0.16 | 60.16 ± 0.18 | 60.12 ± 0.08 | 60.20 ± 0.03 | 60.45 ± 0.09 | 60.88 ± 0.03 | 62.27 ± 0.15 | 65.46 ± 0.40 | 74.15 ± 0.74 | 92.78 ± 0.21 | 99.72 ± 0.03 |
| | No Cue | *Incr.* | 60.91 ± 0.02 | 61.53 ± 0.06 | 61.73 ± 0.12 | 61.99 ± 0.11 | 62.18 ± 0.09 | 62.40 ± 0.09 | 62.59 ± 0.10 | 62.69 ± 0.02 | 62.69 ± 0.05 | 62.43 ± 0.15 | 62.08 ± 0.18 |
| | | *Para.* | 60.90 ± 0.03 | 60.98 ± 0.10 | 60.87 ± 0.05 | 60.85 ± 0.10 | 60.94 ± 0.22 | 60.93 ± 0.09 | 61.07 ± 0.14 | 61.17 ± 0.17 | 61.27 ± 0.24 | 61.23 ± 0.18 | 60.83 ± 0.29 |
| $T_{\text{Color}} \to T_{\text{Color}}$ | $T_{\text{Color}}$ | *Incr.* | 98.50 ± 0.06 | 98.59 ± 0.04 | 98.70 ± 0.05 | 98.76 ± 0.05 | 98.84 ± 0.05 | 98.91 ± 0.05 | 98.96 ± 0.05 | 99.01 ± 0.03 | 99.03 ± 0.02 | 99.07 ± 0.02 | 99.06 ± 0.01 |
| | | *Para.* | 98.49 ± 0.06 | 98.25 ± 0.05 | 97.74 ± 0.12 | 97.24 ± 0.23 | 96.87 ± 0.22 | 96.71 ± 0.25 | 96.87 ± 0.24 | 97.22 ± 0.16 | 97.73 ± 0.14 | 98.17 ± 0.06 | 98.45 ± 0.04 |

Table 3: **Numerical Results for the 'Chunking' Setup.** Displayed for each experiment is the test accuracy (in %), reported as the mean ± standard error across three runs with different random seeds.

| Training | Evaluation | Scenario | Interpolation Coefficient $\alpha$ | | | | | | | | | | |
| --- | --- | --- | --- | --- | --- | --- | --- | --- | --- | --- | --- | --- | --- |
| | | | 0.0 | 0.1 | 0.2 | 0.3 | 0.4 | 0.5 | 0.6 | 0.7 | 0.8 | 0.9 | 1.0 |
| $T_{\text{Color}} \to T_{\text{Noise}}$ | $T_{\text{Color}}$ Cue | *Incr.* | 90.62 ± 0.38 | 89.43 ± 0.40 | 86.98 ± 0.36 | 83.55 ± 0.40 | 79.41 ± 0.56 | 75.00 ± 0.58 | 70.64 ± 0.70 | 66.29 ± 0.78 | 63.63 ± 0.73 | 55.76 ± 0.71 | 53.09 ± 1.04 |
| | | *Para.* | 90.95 ± 0.40 | 87.93 ± 0.30 | 81.71 ± 0.22 | 72.95 ± 0.55 | 62.89 ± 0.67 | 54.67 ± 0.55 | 48.58 ± 0.49 | 46.12 ± 0.44 | 43.71 ± 0.41 | 41.35 ± 0.41 | 39.42 ± 0.61 |
| | $T_{\text{Noise}}$ Cue | *Incr.* | 41.62 ± 0.63 | 43.08 ± 0.61 | 44.49 ± 0.64 | 45.69 ± 0.60 | 46.92 ± 0.61 | 48.60 ± 0.61 | 52.76 ± 0.62 | 60.16 ± 0.64 | 75.12 ± 0.94 | 92.76 ± 0.23 | 99.12 ± 0.05 |
| | | *Para.* | 41.62 ± 0.63 | 42.48 ± 0.61 | 42.82 ± 0.64 | 43.20 ± 0.60 | 43.60 ± 0.61 | 45.25 ± 0.61 | 49.08 ± 0.62 | 56.75 ± 0.64 | 74.12 ± 0.94 | 92.76 ± 0.23 | 99.12 ± 0.05 |
| | CIFAR-100 | *Incr.* | 42.29 ± 0.61 | 43.60 ± 0.52 | 44.49 ± 0.48 | 45.34 ± 0.47 | 46.12 ± 0.46 | 46.53 ± 0.68 | 46.98 ± 0.61 | 47.02 ± 0.48 | 46.87 ± 0.49 | 46.15 ± 0.55 | 45.64 ± 0.59 |
| | | *Para.* | 42.97 ± 0.56 | 43.18 ± 0.50 | 43.23 ± 0.52 | 43.29 ± 0.47 | 43.33 ± 0.48 | 43.15 ± 0.32 | 43.59 ± 0.31 | 43.66 ± 0.31 | 43.59 ± 0.30 | 43.02 ± 0.30 | 41.62 ± 0.40 |
| $T_{\text{Color}} \to T_{\text{Color}}$ | $T_{\text{Color}}$ Cue | *Incr.* | 90.62 ± 0.38 | 91.39 ± 0.39 | 91.93 ± 0.35 | 92.53 ± 0.40 | 93.07 ± 0.40 | 93.60 ± 0.40 | 93.94 ± 0.40 | 94.26 ± 0.40 | 94.56 ± 0.40 | 94.72 ± 0.40 | 94.73 ± 0.40 |
| | | *Para.* | 90.81 ± 0.27 | 90.87 ± 0.28 | 90.83 ± 0.22 | 90.66 ± 0.17 | 90.39 ± 0.16 | 90.24 ± 0.15 | 90.36 ± 0.17 | 90.48 ± 0.17 | 90.68 ± 0.16 | 90.82 ± 0.13 | 90.79 ± 0.04 |

