# OpenReview forum: "Forgetting of task-specific knowledge in model merging-based continual learning"
_TMLR — Withdrawn by Authors_

### Review · Reviewer_e64S · 2026-04-13

**Summary Of Contributions:**

This paper investigates how linear weight interpolation affects different types of knowledge in the context of continual learning. The authors propose a controlled experimental methodology: by superimposing synthetic visual cues onto CIFAR-100 images, they explicitly separate model knowledge into shared components and task-specific components. The paper experiments with a task-specific protocol where endpoint models learn different cues, and a shared protocol where they learn the same cue, for both incremental and parallel training.

The main findings are: (1) shared knowledge is largely preserved or even enhanced during merging; (2) unshared task-specific knowledge degrades rapidly upon interpolation; and (3) merging incrementally trained models consistently outperforms merging models trained in parallel.

Key Strengths:
- The controlled experimental design using synthetic visual cues provides a clean separation between shared and task-specific knowledge, which is difficult to achieve with standard CL benchmarks based on semantic splits.
- The paper is clearly written and the experimental protocol is easy to follow.
- The comparison between incremental and parallel training scenarios is a useful experimental axis that adds some nuance to the analysis.

Key Weaknesses:
- The core findings are largely predictable from prior works on neural network loss landscapes, linear mode connectivity, and model merging. The paper confirms existing intuitions rather than revealing new phenomena.
- The experimental scope is narrow: only CIFAR-100 with a slim ResNet-18, linear interpolation, and two synthetic cue types. This limits the generalizability of the conclusions.
- The practical applicability of the findings is limited. In real-world CL scenarios, the boundary between shared and task-specific knowledge is ambiguous and cannot be cleanly delineated.

**Audience:**

No

**Audience Explanation:**

While model merging for continual learning is an active and relevant research area, the specific findings of this paper are unlikely to provide substantial new insight to the community. The central observation (shared knowledge survives merging while task-specific knowledge does not) is an intuition that is well-established in existing literature (e.g. Goodfellow et al., 2015; Frankle & Carbin, 2019; Neyshabur et al., 2020, all cited in the paper). The finding that models from the same optimization trajectory merge better than independently trained ones is also consistent with classical results on stochastic weight averaging and recent CL work (Izmailov et al., 2018 and Marouf et al., 2024, both cited in the paper). Furthermore, the practical value of these findings is also limited. The paper's controlled setup, while methodologically clean, creates an artificial dichotomy between shared and task-specific knowledge that does not reflect the continuous, overlapping nature of knowledge in realistic settings. The paper does not offer tools, metrics, or methods that would help practitioners decide when to apply model merging in real CL scenarios.

**Broader Impact Concerns:**

N/A.

**Claims And Evidence:**

No

**Claims Explanation:**

The experiments are repeated across multiple seeds with standard error reported, and the two setups (standard and chunking) show consistent trends. In the narrow sense, the experimental evidence presented is accurate and clear.

However, the claims in the paper are made at a general, conceptual level about how model merging affects shared vs. task-specific knowledge, but the evidence comes from an extremely constrained setting. The experiments uses a single small dataset, a single small architecture, the linear model merging method, and artificially injected shortcuts as proxies for task-specific knowledge. The gap between the generality of the claims and the specificity of the evidence is significant. Real task-specific knowledge is encoded in distributed, overlapping representations rather than in easily separable shortcut features.

**Requested Changes:**

There are many possible ways to improve the paper for it to become a contribution of interest to the TMLR audience.

The paper would benefit from deeper analysis, either theoretical grounding or mechanistic investigation into the model's internal representations, that goes beyond confirming expected behavior and instead reveals new understanding of the merging dynamics. For example, analyzing the weight-space geometry between incremental vs. parallel training could provide insight into why incremental training leads to better merging outcomes.

The authors could also broaden the experimental scope. The reliance on a single small-scale setting (CIFAR-100, slim ResNet-18) with only linear interpolation severely limits the generalizability of the conclusions. Testing on additional architectures, datasets, and non-linear model merging methods can better support the paper's general claims. The paper should also validate the findings using a more realistic task structure beyond synthetic visual cues.

---

### Review · Reviewer_2Q3A · 2026-05-12

**Summary Of Contributions:**

This paper studies linear model merging in a controlled CIFAR-100 setup where synthetic label-correlated visual cues are added to images. The authors compare merging models trained with the same cue versus different cues, and compare endpoints from sequential versus parallel training. The main reported finding is that behavior shared by both endpoints is preserved under interpolation, while behavior unique to one endpoint degrades.

**Additional Comments:**

My overall assessment is negative. The paper is readable, but the scientific contribution is too small in its current form. The setup does not adequately represent continual learning, the empirical scale is too limited, and the main takeaway is not surprising. I would encourage the authors to either substantially broaden the experiments or reposition the work as a small controlled study of shortcut-feature behavior under interpolation, rather than as a contribution about model merging for continual learning.

**Audience:**

No

**Audience Explanation:**

The topic is relevant, but the actual findings are too limited and predictable. The paper does not provide a new method, a strong theoretical explanation, or a sufficiently broad empirical study. The main observation, that common behavior survives interpolation better than conflicting endpoint-specific behavior, is intuitive and demonstrated only in a toy cue-based setting. I do not think this provides enough insight for the TMLR audience in its current form.

**Broader Impact Concerns:**

No major broader impact concerns.

**Claims And Evidence:**

No

**Claims Explanation:**

The evidence supports a much narrower claim than the paper makes. The experiments show behavior of linear interpolation between ResNet-18 models trained on CIFAR-100 variants with synthetic visual cues. They do not convincingly support general claims about “task-specific knowledge” in model-merging-based continual learning.

A central issue is that the added “tasks” are not really distinct tasks in the usual continual-learning sense. The class labels, dataset, and objective remain essentially the same. The authors add label-correlated patches that function as shortcut features. This makes the setup closer to controlled data augmentation or spurious-correlation probing than to continual learning over meaningful task sequences. The “shared knowledge” condition is especially weak, since training on TColor -> TColor is basically training twice on the same cue-augmented classification problem.

The experimental scope is also inadequate. The paper uses CIFAR-100, a slim ResNet-18, and mostly two-task sequences. Given the broad framing around continual learning and model merging, there is no clear justification for not extending to larger datasets, more realistic domain/task shifts, larger architectures, and longer task streams. Without these, the claims about continual learning remain under-supported.

**Requested Changes:**

Substantially expand the empirical evaluation beyond CIFAR-100 with synthetic cues. At minimum, include larger datasets, more realistic domain/task shifts, and more than two sequential tasks.

Reframe the paper away from broad continual-learning claims unless the authors add real continual-learning benchmarks. The current “tasks” are cue-augmented versions of the same classification problem, not convincing CL tasks.

Clarify that the results are about synthetic shortcut features, not task-specific knowledge in general. The current terminology overstates what is actually being measured.

Evaluate additional architectures and preferably larger models, such as CLIP. The paper discusses model merging broadly, but the evidence is confined to one small vision architecture.

Compare to stronger merging methods or explain why linear interpolation alone is enough to support the paper’s claims.

---

### Review · Reviewer_3gyH · 2026-05-17

**Summary Of Contributions:**

The paper focuses on how simple linear interpolation and weight averaging affects a CL model merging setting. The authors construct visual tasks on CIFAR-100 by adding class-correlated synthetic cues, and use these cues to distinguish between knowledge that is shared across endpoint models and knowledge that is task-specific or unshared. the main empirical finding is that shared knowledge, including base CIFAR-100 performance and cues common to both endpoints, is largely preserved or sometimes mildly improved under interpolation. On the other hand, unshared cue-specific knowledge degrades rapidly as the interpolation moves away from the corresponding endpoint. They also report that interpolating models obtained along an incremental training trajectory tends to preserve knowledge better than interpolating independently trained task-specialist models.

The main strength of the paper is its clean and interpretable experimental setup: the use of synthetic visual cues provides a simple way to probe shared versus task-specific knowledge under model merging.

However, the contribution is limited in scope and already largely covered by related work (some of it not cited). The paper does not propose a new method, does not provide theoretical analysis, and evaluates only simple linear interpolation on a narrow CIFAR-100/ResNet-style setup.

**Audience:**

No

**Audience Explanation:**

The paper’s findings may be mildly interesting as a controlled demonstration but I do not think they rise to the level of a TMLR contribution. The main intuition (that simple parameter interpolation tends to preserve shared/common components while interfering with unshared, task-specific components) is already well aligned with existing model-merging work, especially prior results on interference, task-vector conflicts, selective forgetting through fusion, and recent continual model-merging methods that explicitly target forgetting. The paper does not substantially change how one would understand or use model merging in continual learning.

In particular, recent related work already motivates and addresses the limitations that this paper illustrates: 1) naive merging can destroy task-specific information 2) sequential/continual merging is nontrivial 3) endpoint training and merging should be co-designed  and 4) feature/representation alignment or interference-aware merging can improve retention.

 I do not see a strong reason why TMLR readers would need this as an additional archival publication

**Claims And Evidence:**

No

**Claims Explanation:**

The paper provides some evidence for its empirical observation, namely that simple linear interpolation preserves shared cue/base knowledge better than unshared cue-specific knowledge. However, the evidence is not sufficiently convincing to support the paper’s general framing about model merging in CL. The experiments are limited to one base dataset, one small architecture family, two synthetic cue types, two-task interpolation, and simple linear merging. The “task-specific knowledge” being forgotten is a shortcut rather than a meaningful task-specific capability, so it is unclear how far the conclusions transfer to realistic continual-learning problems.

The paper also does not compare against stronger or more recent continual model-merging methods that are explicitly designed to mitigate interference and preserve task-specific knowledge. Recent methods such as OPCM, CMM, and OTMF address the same failure mode through projection, optimizer-statistics-based merging, representation alignment, or optimal-transport-based feature alignment, so the paper’s evidence only supports conclusions about naive linear interpolation, not model merging more generally. The reported incremental-vs-parallel advantage is interesting but not mechanistically established: the paper does not provide weight-space distances, representation analyses, loss-barrier measurements, cue-reliance controls, or ablations that would substantiate the proposed explanation. Thus, the evidence supports a narrow diagnostic observation, but not the broader claims or positioning of the paper.

**Requested Changes:**

The most critical adjustment is to substantially revise the related work and positioning. Several recent papers are highly relevant to the submission’s topic but are not cited or discussed. These papers do not merely provide background. They directly address the same phenomena that the submission studies empirically: interference in continual model merging, degradation of task-specific knowledge under naive interpolation, feature/representation mismatch, and the need for endpoint-training or merging procedures designed to preserve task-specific information. Without engaging this work, the paper currently overstates the novelty and importance of its findings. This issue is critical to any possible reconsideration for acceptance.

1. Tang et al., “Merging Models on the Fly Without Retraining: A Sequential Approach to Scalable Continual Model Merging” https://arxiv.org/abs/2501.09522

2. Phan et al., “Toward a Holistic Approach to Continual Model Merging”
https://arxiv.org/abs/2509.23592

3. Pan et al., “Merging without Forgetting: Continual Fusion of Task-Specific Models via Optimal Transport”
https://arxiv.org/abs/2511.19561

More broadly, the paper should make clear that its conclusions apply to simple linear interpolation, not to model merging in general. Recent continual-merging and interference-aware methods are explicitly designed to overcome the task-specific forgetting observed here. Therefore, the paper’s contribution should be reframed as a controlled diagnostic study of one failure mode of naive interpolation, rather than as a broader assessment of model merging for continual learning. This reframing is critical.

A second critical adjustment would be to compare against at least one or two stronger merging methods, or else explicitly narrow the claims. The current experiments only test uniform linear interpolation. Given the existence of projection-based, Fisher/optimizer-statistics-based, OT-based, TIES-style, and representation-alignment-based methods, the paper cannot support general statements about the suitability or limitations of “model merging” for continual learning. If no such comparisons are added, the title, abstract, and conclusions should be revised to emphasize “linear interpolation” or “naive weight averaging.”

---

### Note · Authors · 2026-06-10

**Comment:**

Dear Action Editor and Reviewers,

We have decided to withdraw our submission.

We thank everyone for the time they put into their reviews. We truly appreciate the constructive and fair feedback provided by the entire review team, and apologize for the delay in getting back to you.

**Withdrawal Confirmation:**

I have read and agree with the venue's withdrawal policy on behalf of myself and my co-authors.